# Learning to Contextually Aggregate Multi-Source Supervision for Sequence Labeling

## Abstract

Sequence labeling is a fundamental framework for various natural language processing problems including part-of-speech tagging and named entity recognition. Its performance is largely influenced by the annotation quality and quantity in supervised learning scenarios. In many cases, ground truth labels are costly and time-consuming to collect or even non-existent, while imperfect ones could be easily accessed or transferred from different domains. A typical example is crowd-sourced datasets which have multiple annotations for each sentence which may be noisy or incomplete. Additionally, predictions from multiple source models in transfer learning can be seen as a case of multi-source supervision. In this paper, we propose a novel framework named *Consensus Network* (ConNet) to conduct training with *imperfect annotations from multiple sources*. It learns the representation for every weak supervision source and dynamically aggregates them by a context-aware attention mechanism. Finally, it leads to a model reflecting the consensus among multiple sources. We evaluate the proposed framework in two practical settings of multi-source learning: learning with crowd annotations and unsupervised cross-domain model adaptation. Extensive experimental results show that our model achieves significant improvements over existing methods in both settings. [1]

## 1 Introduction

Sequence labeling is a fundamental framework for various natural language processing (NLP) tasks including part-of-speech (POS) tagging (Ratnaparkhi, 1996), noun phrase chunking (Sang & Buchholz, 2000), word segmentation (Low et al., 2005), and named entity recognition (NER) (Nadeau & Sekine, 2007). Typically, existing methods follow the supervised learning paradigm, and require high-quality annotations. While ground truth labels are expensive and time-consuming, imperfect annotations are much easier to obtain from crowdsourcing (noisy labels) or other domains (out-of-domain). To alleviate or even address the problem of noise and incompleteness, it is important and beneficial to learn from multiple sources.

Specifically, we are interested in two typical application scenarios: 1) **learning with crowd annotations** and 2) **unsupervised cross-domain model adaptation**, which are detailed in Sec. 2 and Sec. 3. The key challenge of learning with multi-source supervision is to aggregate annotators for learning a model without knowing the underlying ground truth label sequences in the target domain. Many attempts have been made in generalizing multiple informative sources to an out-of-domain distribution for a range of tasks, including multi-class classification (Sheshadri & Lease, 2013), object detection (Su et al., 2012) and information extraction (Liu et al., 2017). However, most of the prior works (Nguyen et al., 2017; Wang et al., 2019; Peng & Dredze, 2016; Yang et al., 2017; Chen & Cardie, 2018) choose to use simple heuristic-based methods of aggregating source models for performing on a target corpus without carefully calibration.

Our intuition is mainly from the common phenomenon that each source of supervision has distinct strength in different inputs, and thus they should not keep consistent importance in aggregating supervisions when inferring tag sequences for new inputs. Aggregating multiple sources for a specific input should be a dynamic process depending on the sentence context rather than a fixed way. To

---

[1] Code and data have been uploaded and will be published up to the acceptance of the paper.

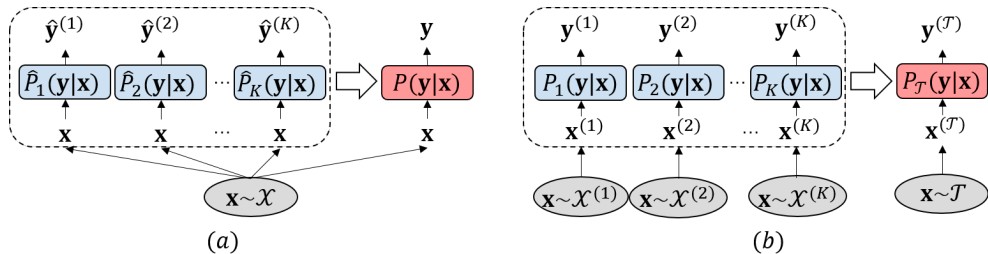

Figure 1: Illustration of the task settings for the two applications in this work: (a) learning consensus model from crowd annotations; (b) unsupervised cross-domain model adaptation.

better model this nature, we need to (1) explicitly model the unique traits of different sources when training and (2) find best suitable sources for generalizing the learned model on unseen sentences.

In this paper, we propose a novel framework, named *Consensus Network* (CONNET), for sequence labeling with multi-source supervisions. We represent the annotation patterns as different biases of annotators over a shared behavior pattern. Both annotator-invariant patterns and annotator-specific biases are modeled in a decoupled way. The first term through sharing part of low-level model parameters in a multi-task learning schema. For learning the biases, we decouple them from the model as the transformations on top-level tagging model parameters, such that they can capture the unique strength of each annotator. With such decoupled source representations, we further learn an attention network for dynamically assigning the best sources for every unseen sentence through composing a transformation that represents the consensus. Extensive experimental results in two scenarios show that our model always outperforms strong baseline methods. CONNET achieves the state-of-the-art performance on real-world crowdsourcing datasets and improve significantly in most unsupervised cross-domain adaptation tasks over existing works. In addition to sequence labeling, it also shows its effectiveness on text classification tasks.

## 2 RELATED WORK

**Neural Sequence Labeling.** Traditional approaches for sequence labeling usually need significant efforts in feature engineering for graphical models like hidden markov models (HMMs) (Rabiner, 1989) and conditional random fields (CRFs) (Lafferty, 2001). Recent research efforts in neural network models have shown that end-to-end learning like convolutional neural networks (CNNs) (Ma & Hovy, 2016b) or bidirectional long short-term memory (BLSTMs) (Lample et al., 2016) can largely eliminate human-crafted features. Together with a final CRF layer, these BLSTM-CRF models have achieved promising performance and are used as our base sequence tagging model in this paper.

**Crowd-sourced Annotation.** Crowd-sourcing has been demonstrated to be an effective way of fulfilling the label consumption of neural models (Guan et al., 2017). It collects annotations with lower costs and a higher speed by non-expert contributors but suffers from some degradation in quality. Dawid & Skene (1979) proposes the pioneering work to aggregate crowd annotations to estimate true labels, and Snow et al. (2008) shows its effectiveness with Amazon's Mechanical Turk system. Later works (Dempster et al., 1977; Dredze et al., 2009; Raykar et al., 2010) focus on Expectation-Maximization (EM) algorithms to jointly learn the model and annotator behavior on classification problems. Recent research shows the strength of multi-task framework in semi-supervised learning (Lan et al., 2018; Clark et al., 2018), and cross-type learning (Wang et al., 2018). Nguyen et al. (2017) and Rodrigues & Pereira (2018) regards crowd annotations as noisy versions of glod labels and constructs crowd components to model annotator-specific bias which were discarded during the inference process.

**Unsupervised Domain Adaptation.** Unsupervised cross-domain adaptation aims to transfer knowledge learned from high-resource domains (source domains) to boost performance on low-resource domains (target domains). Different from supervised adaptation (Lin & Lu, 2018), we assume there is no labels at all for target corpora. Saito et al. (2017) and Ruder & Plank (2018) explored bootstrapping with multi-task tri-training approach. The method is developed for one-to-one domain adaptation and does not model the differences among multiple source domains. Yang & Eisenstein (2015) represents each domain with a vector of metadata domain attributes and uses domain vectors

to train the model to deal with domain shifting. Ghifary et al. (2016) uses an auto-encoder method by jointly training a predictor for source labels, and a decoder to reproduce target input with a shared encoder. The decoder acts as a normalizer to force the model to learn shared knowledge between source and target domains. Adversarial penalty can be applied to the loss function to make models learn domain-invariant feature only (Fernando et al., 2015; Ming Harry Hsu et al., 2015).

# 3 LEARNING WITH MULTI-SOURCE SUPERVISION

We consider the multi-source sequence labeling problem as follows. There are $N$ sources of supervision, each source can be regarded as an imperfect annotator (non-expert human tagger or models trained in related domains). For the $k$-th source data set $S^{(k)} = \{(\mathbf{x}_i^{(k)}, \mathbf{y}_i^{(k)})\}_{i=1}^{m_k}$, we denote its $i$-th sentence as $\mathbf{x}_i^{(k)}$ which is a sequence of tokens: $\mathbf{x}_i^{(k)} = (x_{i,1}^{(k)}, \cdots, x_{i,N}^{(k)})$. The tag sequence of the sentence is thus to be $\mathbf{y}_i^{(k)} = \{y_{i,j}^{(k)}\}$. We define the sentence set of each annotators as $\mathcal{X}^{(k)} = \{\mathbf{x}_i^{(k)}\}_{i=1}^{m_k}$, and the whole training domain as the union of all sentence sets: $\mathcal{X} = \bigcup_{k=1}^{(K)} \mathcal{X}^{(k)}$. The goal of the multi-source learning task is to use such imperfect annotations to train a model for predicting the tag sequence $\mathbf{y}$ for any sentence $\mathbf{x}$ in a target corpus $\mathcal{T}$. Note that the target corpus $\mathcal{T}$ can either share the same distribution with $\mathcal{X}$ (Application I) or be significantly different (Application II). In the following two subsections, we formulate two typical tasks in this problem as shown in Fig. 1.

**Application I: Learning with Crowd Annotations.** When learning with crowd-sourced sequence labeling data, we regard each worker as an imperfect annotator ($S^{(k)}$), who may make mistakes or skip sentences in its annotations. Note that for crowd-sourcing data, different annotators tag subsets of the *same* given dataset ($\mathcal{X}$), and thus we assume there are no input distribution shifts among $\mathcal{X}^{(k)}$. Also, we only test sentences in the same domain such that the distribution in target corpus $\mathcal{T}$ is the same as well. That is, the marginal distribution of target corpus $P_{\mathcal{T}}(\mathbf{x})$ is the same with that for each individual source dataset, *i.e.* $P_{\mathcal{T}}(\mathbf{x}) = P_k(\mathbf{x})$. However, due to imperfectness of the annotations in each source, the $P_k(\mathbf{y}|\mathbf{x})$ has obvious shift from the underlying truth $P(\mathbf{y}|\mathbf{x})$ (illustrated in the top-left part of Fig. 1). The multi-source learning objective here is to learn a model $P_{\mathcal{T}}(\mathbf{y}|\mathbf{x})$ for supporting inference on any new sentences in the same domain.

**Application II: Unsupervised Cross-Domain Model Adaptation.** We assume that we have well-annotated data in some source domains while having no labels in the target domain at all. Following the formulation, we claim that the input distributions $P(\mathbf{x})$ in different source domains $\mathcal{X}^{(k)}$ vary a lot, and fitting those annotations can only generalize to in-domain samples. That is, $P_k(\mathbf{y}|\mathbf{x}) \approx P(\mathbf{y}|\mathbf{x})$ only for $\mathbf{x} \in \mathcal{X}^{(k)}$. For target corpus sentences $\mathbf{x} \in \mathcal{T}$, such a source model $P_k(\mathbf{y}|\mathbf{x})$ again differs from underlying ground truth for the target domain $P_{\mathcal{T}}(\mathbf{y}|\mathbf{x})$ and can be seen as an imperfect annotators. Our objective in this setting is also to jointly model $P_{\mathcal{T}}(\mathbf{y}, \mathbf{x})$ while noticing that there are significant domain shifts between $\mathcal{T}$ and any other $\mathcal{X}^{(k)}$.

# 4 PROPOSED APPROACH: CONSENSUS NETWORK

In this section, we present our two-phase framework CONNET for multi-source sequence labeling. As shown in Figure 2, our proposed framework first uses a multi-task learning schema with a special objective to decouple annotator representations as different parameters of a transformation around CRF layers. This **decoupling phase** (Section 4.2) is for decoupling the model parameters into a set of annotator-invariant model parameters and a set of annotator-specific representations. Secondly, the dynamic **aggregation phase** (Section 4.3) learns to contextually utilize the annotator representations with a lightweight attention mechanism to find the best suitable transformation for each sentence, so that the model can achieve a context-aware consensus among all sources. The inference process is described in Section 4.4.

## 4.1 THE BASE MODEL FOR SEQUENCE LABELING: BLSTM-CRF

Many recent sequence labeling frameworks (Ma & Hovy, 2016a; Misawa et al., 2017) share a very basic structure: a bidirectional LSTM network followed by a CRF tagging layer (i.e. BLSTM-CRF). The BLSTM encodes an input sequence $\mathbf{x} = \{x_1, x_2, \ldots, x_n\}$ into a sequence of hidden state vectors $\mathbf{h}_{1:n}$. The CRF takes as input the hidden state vectors and computes an emission score

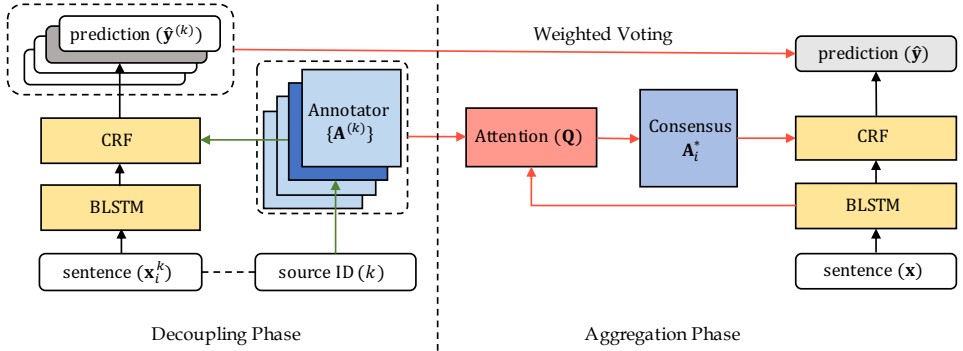

Figure 2: CONNET Overview. The decoupling phase constructs the shared model (yellow) and source-specific matrices (blue). The aggregation phase dynamically combines crowd components into a consensus representation (blue) by a context-aware attention module (red) for each sentence $x$.

matrix $\mathbf{U} \in \mathbb{R}^{n \times L}$ where $L$ is the size of tag set. It also maintains a trainable transition matrix $\mathbf{M} \in \mathbb{R}^{L \times L}$. We can consider $\mathbf{U}_{i,j}$ is the score of labeling the tag with id $j \in \{1, 2, ..., L\}$ for $i^{th}$ word in the input sequence $\mathbf{x}$, and $\mathbf{M}_{i,j}$ means the transition score from $i^{th}$ tag to $j^{th}$.

The CRF further computes the score $s$ for a predicted tag sequence $\mathbf{y} = \{y_1, y_2, ..., y_k\}$ as

$$s(\mathbf{x}, \mathbf{y}) = \sum_{t=1}^{T} (\mathbf{U}_{t,y_t} + \mathbf{M}_{y_{t-1}, y_t}), \tag{1}$$

then tag sequence $\mathbf{y}$ follows the conditional distribution

$$P(\mathbf{y}|\mathbf{x}) = \frac{\exp s(\mathbf{x}, \mathbf{y})}{\sum_{\mathbf{y} \in Y_{\mathbf{x}}} \exp s(\mathbf{x}, \mathbf{y})}. \tag{2}$$

### 4.2 THE DECOUPLING PHASE: LEARNING ANNOTATOR REPRESENTATIONS

For decoupling annotator-specific biases in annotations, we represent them as a transformation on emission scores and transition scores respectively. Specifically, we learn a matrix $\mathbf{A}^{(k)} \in \mathbb{R}^{L \times L}$ for each imperfect annotator $k$ and apply this matrix as transformation on $\mathbf{U}$ and $\mathbf{M}$ as follows:

$$s^{(k)}(\mathbf{x}, \mathbf{y}) = \sum_{t=1}^{T} \left( (\mathbf{U}\mathbf{A}^{(k)})_{t,y_t} + (\mathbf{M}\mathbf{A}^{(k)})_{y_{t-1}, y_t} \right). \tag{3}$$

From this transformation, we can see that the original score function $s$ in Eq. 1 becomes an annotator-specific computation. The original emission and transformation score matrix $\mathbf{U}$ and $\mathbf{M}$ are still shared by all the annotators, while they both are transformed by the matrix $\mathbf{A}^{(k)}$ for $k$-th annotator. While training the model parameters in this phase, we follow a multi-task learning schema. That is, we share the model parameters for BLSTM and CRF (including $\mathbf{W}$, $\mathbf{b}$, $\mathbf{M}$), while updating $\mathbf{A}^{(k)}$ only by examples in $S_k = \{\mathcal{X}^{(k)}, \mathcal{Y}^{(k)}\}$.

The assumption on the annotation representation $\mathbf{A}^{(k)}$ is that it can model the pattern of annotation bias. Each annotator can be seen as a noisy version of the shared model. For the $k$-th annotator, $\mathbf{A}^{(k)}$ models noise from labeling the current word and transferring from the previous label. Specifically, each entry $\mathbf{A}_{i,j}^{(k)}$ captures the probability of mistakenly labeling $i$-th tag to $j$-th tag. In other words, the base sequence labeling model in Sec. 4.1 learns the basic consensus knowledge while annotator-specific components add their understanding to predictions.

### 4.3 THE AGGREGATION PHASE: DYNAMICALLY REACHING CONSENSUS

In the second phase, our proposed network learns a context-aware attention module for a consensus representation supervised by combined predictions on the target data. For each sentence in target data $\mathcal{T}$, we use the model obtained from the decoupling phase to make predictions, and combine predictions using weighted voting. The weight of each source is its normalized $F_1$ score on the

training set. Through weighted voting on such augmented labels over all source sentences $\mathcal{X}$, we can find a good approximation of underlying truth labels.

For better generalization and higher speed, an attention module is trained to estimate the relevance of each source to the target under the supervision of generated labels. Specifically, we compute embedding of each sentence by concatenating the last hidden states of the forward LSTM and the backward LSTM, *i.e.* $\mathbf{h}^{(i)} = [\overrightarrow{\mathbf{h}}_T^{(i)}; \overleftarrow{\mathbf{h}}_0^{(i)}]$. The attention module takes as input the sentence embedding and outputs a normalized weight for each source:

$$\mathbf{q}_i = \mathrm{softmax}(\mathbf{Q}\mathbf{h}^{(i)}), \quad \text{where} \ \ \mathbf{Q} \in \mathbb{R}^{K \times 2d}. \tag{4}$$

where $d$ is the size of each hidden state $\mathbf{h}^{(i)}$. Source-specific matrices $\{\mathbf{A}^{(k)}\}_{k=1}^K$ are then aggregated into a consensus representation $\mathbf{A}_i^*$ for sentence $\mathbf{x}_i \in \mathcal{X}$ by

$$\mathbf{A}_i^* = \sum_{k=1}^K q_{i,k} \mathbf{A}^{(k)}. \tag{5}$$

In this way, the consensus representation contains more information about sources which are more related to the current type of sentence. It also alleviates the contradiction problem among sources, because it could consider multiple sources of different emphasis. Since only an attention model with weight matrix $\mathbf{Q}$ is trained, the amount of computation is relatively small. We assume the sequence labeling model and annotator representations are well-trained in the previous phase. The main objective in this phase is to learn how to select most suitable annotators for the current sentence.

### 4.4 Parameter Learning and Inference of ConNet

ConNet learns parameters through two phases described above. In the first decoupling phase, each training instance from source $S_k$ is used for training the base sequence labeling model and its specific representation $\mathbf{A}^{(k)}$. In the second aggregation phase, we use aggregated predictions from the first phase to learn a lightweight attention module. For each instance in the target corpus $\mathbf{x}_i \in \mathcal{T}$, we calculate its embedding $\mathbf{h}_i$ from BLSTM hidden states. With these sentence embeddings, the context-aware attention module assigns weight $\mathbf{q}_i$ to each source and dynamically aggregates source-specific representations $\{\mathbf{A}^{(k)}\}$ for inferring $\hat{\mathbf{y}}_i$. In the inference process, only the consolidated consensus matrix $\mathbf{A}_i^*$ is applied to the base sequence learning model. In this way, more specialist knowledge helps to deal with more complex instances.

### 4.5 Model Application

The proposed model can be applied to multi-sourcing learning. Here we describe the application in two practical settings: learning with crowd annotations and unsupervised cross-domain model adaptation. In the crowd annotation learning setting, the training data of the same domain is annotated by multiple noisy annotators, and each annotators is treated as a source. In the decoupling phase, the model is trained on noisy annotations, and in the aggregation phase, it is trained with combined predictions on the training set. In the cross-domain setting, the model has access to unlabeled training data of the target domain and clean labeled data of multiple source domains. Each domain is treated as a source. In the decoupling phase, the model is trained on source domains, and in the aggregation phase, the model is trained on combined predictions on the training data of the target domain.

Our framework can also potentially extend to new tasks other than sequence labeling with different encoders. For example, if we use an MLP as encoder, we can transform its hidden representation by multiplying with crowd/consensus matrices. The transformed representation will then used by a decoder to make predictions. We will demonstrate this ability in experiments.

## 5 Experiments

We evaluate ConNet in the two aforementioned settings of multi-source learning: learning with crowd annotations and unsupervised cross-domain model adaptation. Additionally, to demonstrate the generalization of our framework, we also test our method on sequence labeling with transformer encoder in Appendix B and text classification with MLP encoder in Section 5.5.

## 5.1 DATASETS

**Crowd-Annotation Datasets.** We use crowd-annotation datasets based on the 2003 CoNLL shared NER task (Sang & De Meulder, 2003). The real-world datasets, denoted as AMT, are collected by Rodrigues et al. (2014) using Amazon's Mechanical Turk where F1 scores of annotators against the ground truth vary from 17.60% to 89.11%. Since there is no development set in AMT, we also follow Nguyen et al. (2017) to use the AMT training set and CoNLL 2003 development and test sets, denoted as AMTC. Overlapping sentences are removed in the training set, which is ignored in that work. Additionally, we construct two sets of simulated datasets to investigate the quality and quantity of annotators. To simulate the behavior of a non-expert annotator, a CRF model is trained on a small subset of training data and generates predictions on the whole set. Because of the limited size of training data, each model would have a bias to certain patterns.

**Cross-Domain Datasets.** In this setting, we investigate three NLP tasks: POS tagging, NER and text classification. For POS tagging task, we use the GUM portion (Zeldes, 2017) of Universal Dependencies (UD) v2.3 corpus with 17 tags and 7 domains: academic, bio, fiction, news, voyage, wiki, and interview. For NER task, we select the English portion of the OntoNotes v5 corpus (Hovy et al., 2006). The corpus is annotated with 9 named entities with data from 6 domains: broadcast conversation (bc), broadcast news (bn), magazine (mz), newswire (nw), pivot text (pt), telephone conversation (tc), and web (web). Multi-Domain Sentiment Dataset (MDS) v2.0 (Blitzer et al., 2007) is used for text classification, which is built on Amazon reviews from 4 domains: books, dvd, electronics, and kitchen. Since the dataset only contains word frequencies for each review without raw texts, we follow the setting in Chen & Cardie (2018) considering 5,000 most frequent words and use the raw counts as the feature vector for each review.

## 5.2 EXPERIMENT SETUP

For sequence labeling tasks, we follow Liu et al. (2018) to build the BLSTM-CRF architecture as the base model. The dimension of character-level, word-level embeddings and BLSTM hidden layer are set as 30, 100 and 150 respectively. For text classification, We use an MLP with a hidden size of 100 as encoder and a linear classification layer for predicting their labels. The dropout with a probability of 0.5 is applied to the non-recurrent connections for regularization. The network parameters are randomly initialized and updated by stochastic gradient descent (SGD). The learning rate is initialized as 0.015 and decayed by 5% for each epoch. The training process stops early if no improvements in 15 continuous epochs and selects the best model on the development set. For the dataset without a development set, we report the performance on the 50-th epoch. For each experiment, we report the average performance of 3 runs with different random initialization.

## 5.3 COMPARISON WITH BASELINE METHODS

We compare our models with multiple baselines, which can be categorized in two groups: wrapper methods and joint models. To demonstrate the theoretical upper bound of performance, we also train the base model using ground-truth annotations in the target domain (`Gold`).

A wrapper method consists of a label aggregator and a deep learning model. These two components could be combined in two ways: (1) aggregating labels on crowd-sourced training set then feeding the generated labels to a Sequence Labeling Model (`SLM`) (Liu et al., 2017); (2) feeding multi-source data to a Multi-Task Learning (`MTL`) (Wang et al., 2018) model then aggregating multiple predicted labels. We investigate multiple label aggregation strategies. `CONCAT` considers all crowd annotations as gold labels. `MVT` does majority voting on the token level, *i.e.*, the majority of labels $\{\mathbf{y}_{i,j}^k\}$ is selected as the gold label for each token $\mathbf{x}_{i,j}$. `MVS` is conducted on the sequence level, addressing the problem of violating Begin/In/Out (BIO) rules. `DS` (Dawid & Skene, 1979), `HMM` (Nguyen et al., 2017) and `BEA` (Rahimi et al., 2019) induce consensus labels with probability models.

In contrast with wrapper methods, joint models incorporate multi-source data within the structure of sequential taggers and jointly model all of the individual annotators. `CRF-MA` models CRFs with Multiple Annotators by EM algorithm (Rodrigues et al., 2014). Nguyen et al. (2017) augments the LSTM architecture with crowd vectors. These crowd components are element-wise added to the tags scores (`Crowd-Add`) or concatenated to the output of the hidden layer (`Crowd-Cat`). These two methods are the most similar to our extraction phase. We implemented them and got better results

| Methods | AMTC | | | AMT | | |
|---|---|---|---|---|---|---|
| | Precision(%) | Recall(%) | F1-score(%) | Precision(%) | Recall(%) | F1-score(%) |
| CONCAT-SLM | **85.95**(±1.00) | 57.96(±0.26) | 69.23(±0.13) | **91.12**(±0.57) | 55.41(±2.66) | 68.89(±1.92) |
| MVT-SLM | 84.78(±0.66) | 62.50(±1.36) | 71.94(±0.66) | 86.96(±1.22) | 58.07(±0.11) | 69.64(±0.31) |
| MVS-SLM | 84.76(±0.50) | 61.95(±0.32) | 71.57(±0.04) | 86.95(±1.12) | 56.23(±0.01) | 68.30(±0.33) |
| DS-SLM (Nguyen et al., 2017) | 72.30* | 61.17* | 66.27* | - | - | - |
| HMM-SLM (Nguyen et al., 2017) | 76.19* | 66.24* | 70.87* | - | - | - |
| MTL-MVT (Wang et al., 2018) | 81.81(±2.34) | 62.51(±0.28) | 70.87(±1.06) | 88.88(±0.25) | 65.04(±0.80) | 75.10(±0.44) |
| MTL-BEA (Rahimi et al., 2019) | 85.72(±0.66) | 58.28(±0.43) | 69.39(±0.52) | 77.56(±2.23) | 67.23(±0.72) | 72.01(±0.85) |
| CRF-MA (Rodrigues et al., 2014) | - | - | - | 49.40* | **85.60*** | 62.60* |
| Crowd-Add (Nguyen et al., 2017) | 85.81(±1.53) | 62.15(±0.18) | 72.09(±0.42) | 89.74(±0.10) | 64.50(±1.48) | 75.03(±1.02) |
| Crowd-Cat (Nguyen et al., 2017) | 85.02(±0.98) | 62.73(±1.10) | 72.19(±0.37) | 89.72(±0.47) | 63.55(±1.20) | 74.39(±0.98) |
| CL-MW (Rodrigues & Pereira, 2018) | - | - | - | 66.00* | 59.30* | 62.40* |
| CONNET (Ours) | 84.11(±0.71) | **68.61**(±0.03) | **75.57**(±0.27) | 88.77(±0.25) | 72.79(±0.04) | **79.99**(±0.08) |
| Gold (Upper Bound) | 89.48(±0.32) | 89.55(±0.06) | 89.51(±0.21) | 92.12(±0.31) | 91.73(±0.09) | 91.92(±0.21) |

Table 1: Performance on real-world crowd-sourced NER datasets. (* indicates number reported by the paper.)

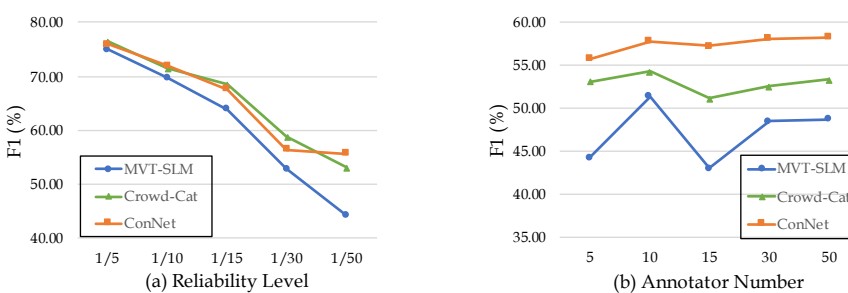

Figure 3: Performance on simulated crowd-sourced NER datasets with (a) 5 annotators on different reliability levels; (b) various numbers of annotators whose reliability $r = 1/50$.

than reported. `CL-MW` applies a crowd layer to a CNN-based deep learning framework (Rodrigues & Pereira, 2018). `Tri-Training` uses bootstrapping with multi-task Tri-Training approach for unsupervised one-to-one domain adaptation (Saito et al., 2017; Ruder & Plank, 2018).

### 5.4 PERFORMANCE ON LEARNING WITH CROWD ANNOTATIONS

Tab. 1 shows the performance of aforementioned methods and our CONNET on two real-world datasets, *i.e.* AMT and AMTC. We can see that CONNET outperforms all other methods on both datasets significantly on $F1$ score, which shows the effectiveness of dealing with noisy annotations for higher-quality labels. Although CONCAT-SLM achieves the highest precision, it suffers from low recall. All existing methods have the high-precision but low-recall problem. One possible reason is that they try to find the latent ground truth and throw away illuminating annotator-specific information. So only simple mentions can be classified with great certainty while difficult mentions fail to be identified without sufficient knowledge. In comparison, CONNET pools information from all annotations and focus on matching knowledge to make predictions. It makes the tagging model be able to identify more mentions and get a higher recall.

It is enlightening to analyze whether the model decides the importance of annotators given a sentence. In Fig. 4 we visualize test F1 score of all annotators on each tag, and attention weights $\mathbf{q}_i$ in Eq. 4 for 4 sampled sentences containing different entity types. Obviously, the 2nd sample sentence with ORG has higher attention weights on 1st, 5th and 33rd annotator who are best at labeling ORG. More details and cases are shown in Appendix A.1. We also investigate multiple variants and conduct ablation study in Appendix A.2.

To analyze the impact of annotator quality, we split the origin train set into $\{5, 10, 15, 30, 50\}$ folds and train a CRF model on each fold whose reliability could be represented as $r = \{1/5, 1/10, 1/15, 1/30, 1/50\}$ because a model with less training data would have stronger bias and less generalization. For each setting, we randomly select 5 trained models as the simulated annotators to annotate the whole training set. When the reliability level of all annotators is too low, *i.e.* $1/50$, only the base model is used for prediction without annotator representations. The models are then trained with simulated annotations. Shown in Fig. 3(a), CONNET achieves significant

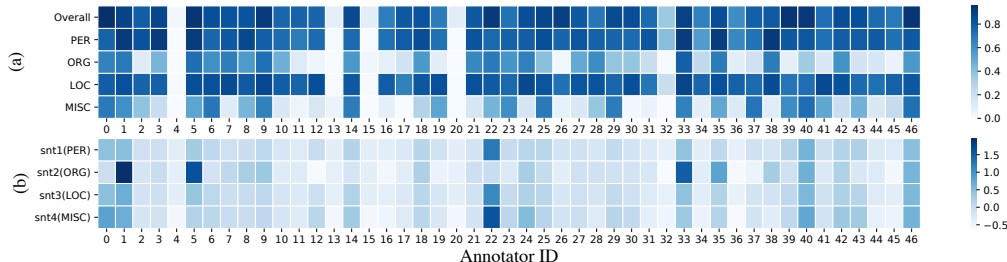

Figure 4: Visualizations of (a) the expertise of annotators; (b) attention weights for sample sentences. More cases and details are described in Appendix A.1.

improvements over `MVT-SLM` and competitive performance as `Crowd-Cat`. Our model shows its effectiveness when annotators are less reliable.

Regarding the influence of annotator quantity, we split the train set into 50 subsets to train 50 models ($r = 1/50$) respectively and randomly select $\{5, 10, 15, 30, 50\}$ models as simulated annotators to annotate the whole training set. The models are then trained with simulated annotations. Fig. 3(b) shows CONNET is superior to baselines and able to well deal with many annotators while there is no obvious relationship between the performance and the number of annotators in baselines. We can see the performance of our model increases as the number of annotators and, regardless of the number of annotators, our method consistently outperforms than other baselines.

## 5.5 PERFORMANCE ON CROSS-DOMAIN ADAPTATION

| Methods | POS Tagging | Named Entity Recognition | Text Classification |
|---|---|---|---|
| CONCAT-SLM | 92.11($\pm$0.07) | 61.24($\pm$0.92) | 79.41($\pm$0.02) |
| MTL-MVT (Wang et al., 2018) | 90.73($\pm$0.29) | 60.44($\pm$0.45) | 77.54($\pm$0.06) |
| MTL-BEA (Rahimi et al., 2019) | 91.71($\pm$0.06) | 52.15($\pm$0.58) | 78.01($\pm$0.28) |
| Crowd-Add (Nguyen et al., 2017) | 91.36($\pm$0.14) | 39.30($\pm$4.44) | 79.30($\pm$9.21) |
| Crowd-Cat (Nguyen et al., 2017) | 91.94($\pm$0.08) | 62.14($\pm$0.89) | 79.54($\pm$0.25) |
| Tri-Training (Ruder & Plank, 2018) | 91.93($\pm$0.01) | 61.67($\pm$0.31) | 80.58($\pm$0.02) |
| CONNET (Ours) | 92.33($\pm$0.17) | **63.32**($\pm$0.81) | **81.55**($\pm$0.04) |
| Gold (Upper Bound) | 92.88($\pm$0.14) | 68.61($\pm$0.64) | 83.22($\pm$0.19) |

Table 2: **Performance on cross-domain adaptation.** The average score for all domains is reported for each task. The best score in each column that is significantly ($p < 0.05$) better than the second-best is marked **bold**, while those are better but not significantly are underlined. Detailed results can be found in Appendix A.3.

The average performance of each method on each task is shown in Tab. 2. More detailed results on each target domain can be found in Appendix A.3. We report the accuracy for POS tagging and text classification, and report the chunk-level F1 score for NER. We can see that CONNET achieves the highest average score in all tasks. `MTL-MVT` is similar to our decoupling phase without the attention module, which performs much worse. It shows that naively doing unweighted voting does not work well. The attention can be viewed as implicitly doing weighted voting on the feature level. This demonstrates the importance of having such a module to assign weights to all domains based on the input sentence. The Tri-Training model trained on the concatenated training data from all sources performs worse than CONNET, which suggests that it is important to have a multi-task structure to model the difference among domains. We analyze the OntoNotes dataset to show that the attention scores generated by the model are meaningful. Details can be found in Appendix A.4.

## 6 CONCLUSION

In this paper, we present CONNET for learning a sequence tagger from multi-source supervision. It could be applied in two practical scenarios: learning with crowd annotations and cross-domain adaptation. In contrast to prior works, CONNET learns fine-grained representations of each source which are further dynamically aggregated for every unseen sentence in the target data. Experiments show that our model is superior to previous crowd-sourcing and unsupervised domain adaptation sequence labeling models. The proposed learning framework also shows promising results on other NLP tasks like text classification.

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

# A   ANALYSIS OF CONNET WITH BLSTM ENCODER

## A.1   CASE STUDY ON LEARNING WITH CROWD ANNOTATIONS

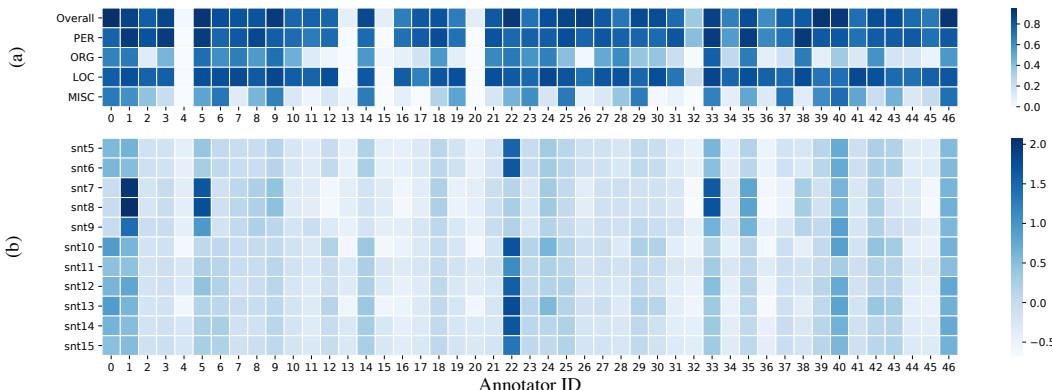

Figure 5: Visualizations of (a) the expertise of annotators; (b) attention weights for additional sample sentences to Fig. 4. Details of samples are described in Tab. 3.

| 1 | Defender [PER Hassan Abbas] rose to intercept a long ball into the area in the 84th minute but only managed to divert it into the top corner of [PER Bitar] 's goal . |
|---|---|
| 2 | [ORG Plymouth] 4 [ORG Exeter] 1 |
| 3 | Hosts [LOC UAE] play [LOC Kuwait] and [LOC South Korea] take on [LOC Indonesia] on Saturday in Group A matches . |
| 4 | The former [MISC Soviet] republic was playing in an [MISC Asian Cup] finals tie for the first time . |
| 5 | [PER Bitar] pulled off fine saves whenever they did . |
| 6 | [PER Coste] said he had approached the player two months ago about a comeback . |
| 7 | [ORG Goias] 1 [ORG Gremio] 3 |
| 8 | [ORG Portuguesa] 1 [ORG Atletico Mineiro] 0 |
| 9 | [LOC Melbourne] 1996-12-06 |
| 10 | On Friday for their friendly against [LOC Scotland] at [LOC Murrayfield] more than a year after the 30-year-old wing announced he was retiring following differences over selection . |
| 11 | Scoreboard in the [MISC World Series] |
| 12 | Cricket - [MISC Sheffield Shield] score . |
| 13 | " He ended the [MISC World Cup] on the wrong note , " [PER Coste] said . |
| 14 | Soccer - [ORG Leeds] ' [PER Bowyer] fined for part in fast-food fracas . |
| 15 | [ORG Rugby Union] - [PER Cuttitta] back for [LOC Italy] after a year . |
| 16 | [LOC Australia] gave [PER Brian Lara] another reason to be miserable when they beat [LOC West Indies] by five wickets in the opening [MISC World Series] limited overs match on Friday . |

Table 3: Sample instances in Fig. 4 and Fig. 5 with NER annotations including PER (red), ORG (blue), LOC (violet) and MISC (orange).

To better understand the effect and benefit of CONNET, we do some case study on AMTC real-world dataset with 47 annotators. We look into some more instances to investigate the ability of attention module to find right annotators in Fig. 5 and Tab. 3. Sentence 1-12 contains a specific entity type respectively while 13-16 contains multiple different entities. Compared with expertise of annotators, we can see that the attention module would give more weight on annotators who have competitive performance and preference on the included entity type. Although top selected annotators for ORG has relatively lower expertise on ORG than PER and LOC, they are actually the top three annotators with highest expertise on ORG.

## A.2   ABLATION STUDY ON CONNET

To investigate the effectiveness of our proposed method, multiple variants of the decoupling phase and aggregation phase are evaluated on the AMT dataset, shown in Fig. 6(a).

**Decoupling Phase** We tried three different approaches to incorporate source-specific representation in the decoupling phase (DP). CRF means the traditional approach without annotator matrices as Eq. 1 while DP(1+2) is described in Eq. 3. Similarly, DP(1) only applies source representations $\mathbf{A}^{(k)}$ to the emission score $\mathbf{U}$ while DP(2) only transfers the transition matrix $\mathbf{M}$. They directly use latent representations $\mathbf{U}$ and $\mathbf{M}$ in the decoupling phase (not augmented by $\mathbf{A}^{(k)}$) for prediction. We can observe from the result that both variants that model the annotator-specific information can improve the result. The underlying model keeps more consensus knowledge if we extract annotator-specific bias on sentence encoding and label transition.

**Aggregation Phase** We compare four methods of generating supervision targets in the aggregation phase (AP). OMV uses majority voting of original annotations as targets for training, while PMV substitutes them with model prediction learned from DP. AMV extends the model by using all prediction, while AWV uses majority voting weighted by each annotator's training $F1$ score. The results show the effectiveness of AWV, which could well approximate the ground truth and supervise the attention module to estimate the expertise of annotator on the current sentence. We can also infer labels on the test set by conducting AWV on predictions of the underlying model with each annotator-specific components. But it leads to heavy computation-consuming and unsatisfying performance, whose test $F1$ score is $77.35(\pm 0.08)$. To investigate the importance of extracted source-specific components, we trained a traditional BLSTM-CRF model by the supervision of the same AMV labels. Its result is $78.93(\pm 0.13)$, which is lower than our CONNET.

These experiments convey four messages: (1) Using learned model prediction is better than directly using original annotations as its target; (2) It is desirable to use all model predictions instead of just confine each annotator on their data set. This means that the learned representation of individual annotator can generalize on data it didn't annotate. (3) Weighting aggregating models by their performance is beneficial to overall accuracy. (4) The attention model could learn to estimate annotators' ability based on the context.

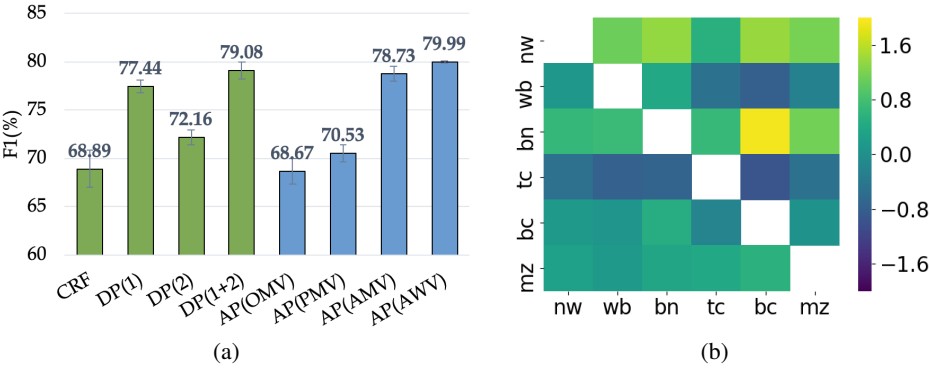

Figure 6: (a) Performance of CONNET variants for ablation study; (b) Heatmap of averaged attention scores from each source domain to each target domain.

## A.3   DETAILED RESULTS FOR CROSS-DOMAIN ADAPTATION

The results of each task on each domain are shown in Tab. 4. We can see that CONNET performs the best on most of the domains and achieves the highest average score for all tasks.

## A.4   MULTI-DOMAIN: ANALYSIS

We analyzed the attention scores generated by the attention module on the OntoNotes dataset. For each sentence in the target domain we collected the attention score of each source domain, and finally

the attention scores are averaged for each source-target pair. Fig. 6(b) shows all the source-to-target average attention scores. We can see that some domains can contribute to other related domains. For example, bn (broadcast news) and nw (newswire) are both about news and they contribute to each other; bn and bc (broadcast conversation) are both broadcast and bn contributes to bc; bn and nw both contributes to mz (magzine) probably because they are all about news; wb (web) and tc (telephone conversation) almost make no positive contribution to any other, which is reasonable because they are informal texts compared to others and they are not necessarily related to the other. Overall the attention scores can make some sense to human, and it suggests that the attention is meaningful and contributing to the model.

| Task & Corpus | Multi-Domain POS Tagging: Universal Dependencies v2.3 - GUM | | | | | | | |
|---|---|---|---|---|---|---|---|---|
| Target Domain | academic | bio | fiction | news | voyage | wiki | interview | AVG $Acc(\%)$ |
| CONCAT | 92.68 | 92.12 | 93.05 | 90.79 | 92.38 | 92.32 | 91.44 | 92.11($\pm$0.07) |
| MTL-MVT (Wang et al., 2018) | 92.42 | 90.59 | 91.16 | 89.69 | 90.75 | 90.29 | 90.21 | 90.73($\pm$0.29) |
| MTL-BEA (Rahimi et al., 2019) | 92.87 | 91.88 | 91.90 | 91.03 | 91.67 | 91.31 | 91.29 | 91.71($\pm$0.06) |
| Crowd-Add (Nguyen et al., 2017) | 92.58 | 91.91 | 91.50 | 90.73 | 91.74 | 90.47 | 90.61 | 91.36($\pm$0.14) |
| Crowd-Cat (Nguyen et al., 2017) | 92.71 | 91.71 | 92.48 | 91.15 | 92.35 | 91.97 | 91.22 | 91.94($\pm$0.08) |
| Tri-Training (Ruder & Plank, 2018) | 92.84 | 92.15 | 92.51 | 91.40 | 92.35 | 91.29 | 91.00 | 91.93($\pm$0.01) |
| CONNET | 92.97 | 92.25 | 93.15 | 91.06 | 92.52 | **92.74** | 91.66 | 92.33($\pm$0.17) |
| Gold (Upper Bound) | 92.64 | 93.10 | 93.15 | 91.33 | 93.09 | 94.67 | 92.20 | 92.88($\pm$0.14) |
| Task & Corpus | Multi-Domain NER: OntoNotes v5.0 - English | | | | | | | |
| Target Domain | nw | wb | bn | tc | bc | mz | | AVG $F_1(\%)$ |
| CONCAT | 68.23 | 32.96 | 77.25 | 53.66 | 72.74 | 62.61 | | 61.24($\pm$0.92) |
| MTL-MVT (Wang et al., 2018) | 65.74 | 33.25 | 76.80 | 53.16 | 69.77 | 63.91 | | 60.44($\pm$0.45) |
| MTL-BEA (Rahimi et al., 2019) | 58.33 | 32.62 | 72.47 | 47.83 | 48.99 | 52.68 | | 52.15($\pm$0.58) |
| Crowd-Add (Nguyen et al., 2017) | 45.76 | 32.51 | 50.01 | 26.47 | 52.94 | 28.12 | | 39.30($\pm$4.44) |
| Crowd-Cat (Nguyen et al., 2017) | 68.95 | 32.61 | 78.07 | 53.41 | **74.22** | 65.55 | | 62.14($\pm$0.89) |
| Tri-Training (Ruder & Plank, 2018) | 69.68 | 33.41 | 79.62 | 47.91 | 70.85 | 68.53 | | 61.67($\pm$0.31) |
| CONNET | **71.31** | **34.06** | 79.66 | 52.72 | 71.47 | **70.71** | | **63.32**($\pm$0.81) |
| Gold (Upper Bound) | 84.70 | 46.98 | 83.77 | 52.57 | 73.05 | 70.58 | | 68.61($\pm$0.64) |
| Task & Corpus | Multi-Domain Text Classification: MDS | | | | | | | |
| Target Domain | books | | dvd | electronics | | kitchen | | AVG $Acc(\%)$ |
| CONCAT | 75.68 | | 77.02 | 81.87 | | 83.07 | | 79.41($\pm$0.02) |
| MTL-MVT (Wang et al., 2018) | 74.92 | | 74.43 | 79.33 | | 81.47 | | 77.54($\pm$0.06) |
| MTL-BEA (Rahimi et al., 2019) | 74.88 | | 74.60 | 79.73 | | 82.82 | | 78.01($\pm$0.28) |
| Crowd-Add (Nguyen et al., 2017) | 75.72 | | 77.35 | 81.25 | | 82.90 | | 79.30($\pm$9.21) |
| Crowd-Cat (Nguyen et al., 2017) | 76.45 | | 77.37 | 81.22 | | 83.12 | | 79.54($\pm$0.25) |
| Tri-Training (Ruder & Plank, 2018) | 78.75 | | 78.45 | 81.95 | | 83.17 | | 80.58($\pm$0.02) |
| CONNET | 77.58 | | **81.06** | **84.12** | | 83.45 | | **81.55**($\pm$0.04) |
| Gold (Upper Bound) | 78.78 | | 82.11 | 86.21 | | 85.76 | | 83.22($\pm$0.19) |

Table 4: **Performance on cross-domain adaptation.** The best score (except the Gold) in each column that is significantly ($p < 0.05$) better than the second best is marked **bold**, while those are better but not significantly are underlined. * means the results from the paper.

## B    RESULT OF CONNET WITH TRANSFORMER ENCODER

| Methods | AMTC | | | UD |
|---|---|---|---|---|
| | Precision(%) | Recall(%) | F1-score(%) | Accuracy(%) |
| MVT-SLM | 72.21($\pm$1.63) | 51.72($\pm$3.58) | 60.21($\pm$1.87) | 87.23($\pm$0.51) |
| Crowd-Add (Nguyen et al., 2017) | 75.32($\pm$1.41) | 50.80($\pm$0.30) | 60.68($\pm$0.67) | 88.20($\pm$0.36) |
| CONNET (Ours) | **76.86**($\pm$0.33) | **56.43**($\pm$3.32) | **65.05**($\pm$2.32) | **89.27**($\pm$0.31) |
| Gold (Upper Bound) | 81.24($\pm$1.25) | 80.52($\pm$0.37) | 80.87($\pm$0.79) | 90.45($\pm$0.71) |

Table 5: Performance of methods with Transformer-CRF as the base model on crowd-annotation NER dataset AMTC and cross-domain POS dataset UD.

To demonstrate the generalization of our framework, we re-implement CONNET and some baselines (`MTV-SLM`, `Crowd-Add`, `Gold`) with Transformer-CRF as the base model. Specifically, the base model takes Transformer as the encoder for CRF, which has shown its effectiveness in many NLP tasks (Vaswani et al., 2017; Devlin et al., 2018). Transformer models sequences with self-attention and eliminates all recurrence. Following the experimental settings from Vaswani et al. (2017), we set the number of heads for multi-head attention as $8$, the dimension of keys and values as $64$, and the hidden size of the feed-forward layers as $1024$. We conduct experiments with crowd-annotation dataset AMTC on NER task and cross-domain dataset UD on POS task, which are described in Section 5.1. Results are shown in Table 5. We can see our model outperforms over other baselines in both tasks and applications.

