# OpenReview forum: "Learning to Contextually Aggregate Multi-Source Supervision for Sequence Labeling"
_ICLR.cc/2020/Conference — Reject_

### Official Review · AnonReviewer3 · 2019-10-21
**Official Blind Review #3**

**Rating:** 6

**Review:**


This paper proposes ConNet, a new label aggregation method for sequence labeling tasks, including crowd-annotation and cross-domain model adaptation. The model consists of a decoupling phase, which learns annotator-specific transforming matrices A, and an aggregation phase with an attention module. Extensive experimental results demonstrate the superiority of the proposed model over baselines.
The paper is generally well-written and easy to follow, and the results seem convincing, so I think it can inspire other works on this topic. My main concern on the paper is its generalization. Crowdsourcing usually involves lots of annotators, and some of them only give very few labels. In these situations, the proposed model introduces lots of new parameters (A, Q), which may cause difficulty during training. So the positive results in Fig 3(b) are very important to dispel my worry, which requires more explanation.
Below are some detailed questions:
-      How do you calculate the sentence embeddings h_i during training?
-      Have you introduced some regularization terms on A and Q?
-      What are the most important hyper-parameters for this model, and how to tune?
-      Can this method be extended to new tasks other than sequence labeling?
-      Can you compare your method with the aggregation method used in on-the-job learning paper [1]?
-      92.33 in tab 2 shouldn’t be bold.

[1] Werling K , Chaganty A , Liang P , et al. On-the-Job Learning with Bayesian Decision Theory[J]. Computer Science, 2015.


**Experience Assessment:**

I have published one or two papers in this area.

**Review Assessment: Checking Correctness Of Derivations And Theory:**

I assessed the sensibility of the derivations and theory.

**Review Assessment: Checking Correctness Of Experiments:**

I did not assess the experiments.

**Review Assessment: Thoroughness In Paper Reading:**

I read the paper at least twice and used my best judgement in assessing the paper.

---

> ### Author Response · Authors · 2019-11-11
> **Thank you for the feedback**
>
> Thank you for your comments!
>
> -------------------------------
> Q1: My main concern on the paper is its generalization. Crowdsourcing usually involves lots of annotators, and some of them only give very few labels. In these situations, the proposed model introduces lots of new parameters (A, Q), which may cause difficulty during training. So the positive results in Fig 3(b) are very important to dispel my worry, which requires more explanation.
>
> A1: Regarding Fig 3(b), each annotator model (CRF) is trained with only 1/50 of the training data (reliability=1/50) to simulate noisy annotators. The CRFs are used to make predictions on the entire training data which are then used as noisy annotations. The figure shows that (1) the performance of our model increases as the number of annotators and (2) regardless of the number of annotators, our method consistently outperforms than other baselines.
>
> We compare our method and all baselines on performance and the number of parameters. The baseline models we used for comparison can be classified into two groups. The first group is those that treat all sources as a single source (concatenates training data from all sources): CONCAT-SLM, CRF-MA, Tri-Training. Compared to this group, our model introduces A and Q with additional parameters. But our model outperforms them, suggesting that it’s beneficial to model each source separately. The second group is those that model each source specifically: MVT/MVS/DS/HMM-SLM, MTL-MVT/BEA, Crowd-Add/Cat, CL-MW. Compared to this group, our model only introduces Q with additional parameters and achieves better performance.
>
> We argue that the number of parameters in A and Q is not very significant. The sizes of new parameters A and Q are linear to the number of annotators. Assume #tag = 10 (e.g., CoNLL 2003 has 9 tags), #annotator = 100, and #hidden_units = 100, the number of parameters in A is (#annotator x #tag x #tag = 10k) and in Q is (#annotator x 2#hidden_units = 20k). But a normal 300-to-100 BLSTM has around 300k parameters.
>
> -------------------------------
> Q2: How do you calculate the sentence embeddings h_i during training?
>
> A2: We concatenate the last hidden states of the forward LSTM and the backward LSTM as the sentence embedding
>
> -------------------------------
> Q3: Have you introduced some regularization terms on A and Q?
>
> A3: No, we don’t have a motivation to regularize A and Q.
>
> -------------------------------
> Q4: What are the most important hyper-parameters for this model, and how to tune?
>
> A4: We did not include additional hyper-parameters to control the size of the crowd matrices or the attention module. Therefore all hyper-parameters are related to the base model BLSTM-CRF (and MLP for text classification), for example, pre-trained word embeddings, character representation, and hidden size of the encoder [1].
> [1] Nils Reimers and Iryna Gurevych. 2017. Optimal Hyperparameters for Deep LSTM-Networks for Sequence Labeling Tasks.
>
> -------------------------------
> Q5: Can this method be extended to new tasks other than sequence labeling?
>
> A5: Yes, we have tested our method on text classification with an MLP as the encoder (see cross-domain experiments, we emphasized it more in the new version). Actually the proposed method can apply to any encoders, but we found it performs better with CRF.
>
> -------------------------------
> Q6: Can you compare your method with the aggregation method used in on-the-job learning paper [1]?
>
> A6: The paper does not focus on aggregation and only has a few words describing it: “$p_θ(y|x)$ is the prediction model… The CRF model $p_θ(y|x)$ is learned based on all actual responses (not simulated ones) using AdaGrad”. Based on these explanations it looks the same as concatenating data from all sources to train a single model (CONCAT-SLM). We are trying to look more into it and see if we can reimplement the method.
>
> -------------------------------
> Q7: 92.33 in tab 2 shouldn’t be bold.
>
> A7: Nice catch! Thanks for pointing out and we will do another round of editing to improve the presentation.

---

### Official Review · AnonReviewer1 · 2019-10-23
**Official Blind Review #1**

**Rating:** 6

**Review:**


## Updated review

I have read the rebuttals. The new version of the paper is clearer and the new baseline experiments are a good addition.

## Original review

This paper presents an approach to train a neural networks-based model for sequence modelling using labels from different sources. The proposed approach explicitly models the annotator and uses an attention model to select the best aggregation method. The model is used on two scenarios: learning with crowd annotation and cross-domain adaptation. For the first scenario, noisy annotators are simulated with models trained on subsets of the data, the proposed model is compared with related works and is shown to achieve the highest f-1 score. For the second scenario, different domains in three NLP tasks are used and the model is shown to yield the best performance.

I think this paper should be accepted, for the following reasons:
- The approach is novel as far as I can tell, and the approach of learning to aggregate labels is significant, as it could also be applied to tasks where inter-annotators agreement is a problem.
- The experiments are convincing and show the potential of the proposed approach.
- The comparison with related works is thorough.

Detailed comments
- I don't understand the notion of "normalized expertise" in Section 5.4, can the authors briefly describe it in the paper ?
- The paper is not easy to read, for instance the first paragraph of Section 5 contains critical information to understand the experiments, maybe it should be moved the Section 4 and developed more, typically in two subsections "Application to crowd annotation" and "Application to cross-domain" for example.
- Typos:
    - Section 2, 3rd paragraph "for traget corpora" -> "target"
    - Same paragraph: "Yang & Eisenstien (2018) represented" -> "represents" to be consistent
    - Section 4.1: "BiLSTM-CRF" -> "BLSTM-CRF"


**Experience Assessment:**

I do not know much about this area.

**Review Assessment: Checking Correctness Of Derivations And Theory:**

I assessed the sensibility of the derivations and theory.

**Review Assessment: Checking Correctness Of Experiments:**

I assessed the sensibility of the experiments.

**Review Assessment: Thoroughness In Paper Reading:**

I read the paper at least twice and used my best judgement in assessing the paper.

---

> ### Author Response · Authors · 2019-11-11
> **Thank you for the feedback**
>
> Thank you for your comments and suggestions!
>
> -------------------------------
> Q1: What does the notion of "normalized expertise" in Section 5.4 mean?
>
> A1: "Normalized expertise" refers to the F1 score on the test set, which represents the expertise level of the annotator. To make the representation clearer, we change the terminology as "test F1 score" in the revision.
>
> -------------------------------
> Q2: The paper is not easy to read.
>
> A2: We appreciate your suggestions on the paper structure and typos. We have rearranged Section 4-5 and fix all the typos in the new version to make it clearer and more readable. Specifically, Section 4.5 is added for model applications: learning with crowd annotations and unsupervised cross-domain model adaptation while the first paragraph of Section 5 has been simplified.

---

### Official Review · AnonReviewer2 · 2019-11-04
**Official Blind Review #2**

**Rating:** 3

**Review:**

This work proposes a method to learn how to aggregate weak supervision sources in the context of sequence labeling. In particular, the model has two main steps: i) it learns a source dependent transformation and ii) it learns a mechanism to combine them.

In general, the paper is well organized but it is not easy to follow. It was not clear to me how the authors combine the source dependent representation in Section 4.2 with the aggregation phase presented in Section 4.3. In particular, it is confusing to me how the model combines during the training Eq. 3 and Eq. 5. Similarly, it is not clear to me how in Eq. 6 the model can calculate attention coefficients based only on information from the sentence embedding (h^(i)).

This work assumes a BLSTM-CRF architecture as a baseline, but it does not explore alternative approaches, such as transformer.

In terms of the experiments, authors evaluate the resulting model using two application settings: combining noisy crowd annotations (AMT) and unsupervised cross-domain model adaptation. Results are encouraging, the proposed method is able to outperforms several recent works in terms of F1 metric for the case of AMT and accuracy for the case of cross-domain adaptation. Qualitative results also shows reasonable performance. The supplemental material also include an ablation study.

In summary, the proposed method is interesting and results seem to be encouraging, however, there are parts of the proposed method that are not clear to me. I rate the paper as weak reject.

**Experience Assessment:**

I do not know much about this area.

**Review Assessment: Checking Correctness Of Derivations And Theory:**

I assessed the sensibility of the derivations and theory.

**Review Assessment: Checking Correctness Of Experiments:**

I assessed the sensibility of the experiments.

**Review Assessment: Thoroughness In Paper Reading:**

I made a quick assessment of this paper.

---

> ### Author Response · Authors · 2019-11-11
> **Thank you for the feedback**
>
> Thank you for your comments and feedback! We have rewritten and rearranged the section in the new version to make it clearer and more readable.
>
> -------------------------------
> Q1: How to combine the source dependent representation in Section 4.2 with the aggregation phase presented in Section 4.3? In particular, how the model combines during the training Eq. 3 and Eq. 5? Similarly, how in Eq. 6 the model can calculate attention coefficients based only on information from the sentence embedding (h^(i))?
>
> A1: In the decoupling phase, we train a shared BLSTM-CRF model with source-specific matrices applied individually to the CRF (Eq.3). This is essentially a multi-task model. In this phase, the model is trained with the entire training data from all sources.
>
> In the dynamic aggregation phase, we train only an attention module which takes as input the sentence encoding and outputs an attention score for each crowd matrix (Eq.4, where Q is trainable parameters). We then take a weighted average (weighted by attention scores) of all the crowd matrices to form the Consensus Matrix (Eq.5) which is to be applied to the CRF module the same way as the crowd matrices (same as Eq.3). In this phase, the attention module is trained on the target training data with “pseudo labels” (weighted voting of predictions of the multi-task model obtained in the decoupling phase, detailed in Appendix A.2).
>
> The motivation of having such an attention module is that we want to take advantage of the source-specific knowledge in addition to the shared knowledge for the target data. And the target may be more related to some sources than other sources. So the attention module is used to determine the weight of each source to the target and aggregate source-specific information into the Consensus Matrix, which is then applied to the CRF. In cross-domain experiments, the attention module is trained on the target data with “pseudo labels” to be familiar with the style of the domain.
>
> We also updated Section 4.3 and 4.5 for better clarification. Thank you!
>
> -------------------------------
> Q2: This work assumes a BLSTM-CRF architecture as a baseline, but it does not explore alternative approaches, such as a transformer.
>
> A2. Regarding the base model, we considered that BLSTM-CRF is generally used as the base model for sequence tagging [1, 2, 3]. In addition to BLSTM-CRF, we also explored our method with MLP encoder for text classification (see cross-domain experiments, we emphasized it more in the new version). We will conduct more experiments with the transformer as an encoder to demonstrate the effectiveness of our method.
>
> [1] Guillaume Lample, Miguel Ballesteros, Sandeep Subramanian, Kazuya Kawakami, and Chris Dyer.  Neural architectures for named entity recognition.
> [2] Liyuan Liu, Xiang Ren, Jingbo Shang, Jian Peng, and Jiawei Han.  Efficient Contextualized Representation: Language Model Pruning for Sequence Labeling
> [3] Hang Yan, Bocao Deng, Xiaonan Li, and Xipeng Qiu.  TENER: Adapting Transformer Encoder for Name Entity Recognition

---

> ### Author Response · Authors · 2019-11-14
> **Experimental results with a transformer**
>
> Good news! We re-implement our method and some baselines (MTV-SLM, Crowd-Add, Gold) with a transformer encoder, and conduct experiments with crowd-annotation dataset AMTC on NER task and cross-domain dataset UD on POS task. Our model outperforms over other baselines in both tasks and applications as the following table shows. The experiments are mentioned in the first paragraph of section 5. Details and results can be found in Appendix B of the updated paper.
>
> method                        |    AMTC (F1%)    |     UD (acc%)
> MTV-SLM                     |   60.21($\pm$1.87)   |    87.23($\pm$0.51)
> Crowd-Add                  |   60.68($\pm$0.67)   |    88.20($\pm$0.36)
> ConNet (Ours)            |   65.05($\pm$2.32)   |    89.27($\pm$0.31)
> Gold (Upper bound)  |   80.87($\pm$0.79)   |    90.45($\pm$0.71)

---

### Author Response · Authors · 2019-11-11
**Paper Revised and Additional Experiments Running**

Thank all reviewers for careful reviews and helpful comments. We have taken each comment seriously and attempted to address every concern.

We updated the paper as follows:
- Rearranged section 4 and 5, as suggested by reviewer #1;
- Improved clarity in section 4.3, as reviewer #2 concerns;
- Explained more about model parameters and Fig 3(b), as reviewer #3 suggests;
- Emphasized more on the text classification task which may be missed by readers before.

We are also re-running some experiments with Transformer as substitution of BLSTM to demonstrate the effectiveness of our method with different encoders, which is in response to reviewer #2. We will report back the performance comparison in one or two days.

---

### Author Response · Authors · 2019-11-14
**New experiments with a transformer encoder**

In response to reviewer #2, we re-implement our method and some baselines with a transformer encoder. New experiments are conducted with the crowd-annotation NER dataset and cross-domain POS dataset. The results show that our model outperforms over other baselines in both tasks and applications.

The new version mentions it in the first paragraph of section 5 and describes more details and results in Appendix B.

---

### Decision · Program_Chairs · 2019-12-19

**Decision:**

Reject

**Comment:**

While the revised paper was better and improved the reviewers assessment of the work, the paper is just below the threshold for acceptance. The authors are strongly encouraged to continue this work.